# Underwater Holothurian Target-Detection Algorithm Based on Improved CenterNet and Scene Feature Fusion

**DOI:** 10.3390/s22197204

**Published:** 2022-09-22

**Authors:** Yanling Han, Liang Chen, Yu Luo, Hong Ai, Zhonghua Hong, Zhenling Ma, Jing Wang, Ruyan Zhou, Yun Zhang

**Affiliations:** 1College of Information Technology, Shanghai Ocean University, Shanghai 201306, China; 2Guangdong Feida Transportation Engineering Co., Ltd., Guangzhou 510663, China

**Keywords:** holothurian, underwater target detection, CenterNet, transformer, scene feature fusion, context information

## Abstract

Aiming at the common problems, such as noise pollution, low contrast, and color distortion in underwater images, and the characteristics of holothurian recognition, such as morphological ambiguity, high similarity with the background, and coexistence of special ecological scenes, this paper proposes an underwater holothurian target-detection algorithm (FA-CenterNet), based on improved CenterNet and scene feature fusion. First, to reduce the model’s occupancy of embedded device resources, we use EfficientNet-B3 as the backbone network to reduce the model’s Params and FLOPs. At the same time, EfficientNet-B3 increases the depth and width of the model, which improves the accuracy of the model. Then, we design an effective FPT (feature pyramid transformer) combination module to fully focus and mine the information on holothurian ecological scenarios of different scales and spaces (e.g., holothurian spines, reefs, and waterweeds are often present in the same scenario as holothurians). The co-existing scene information can be used as auxiliary features to detect holothurians, which can improve the detection ability of fuzzy and small-sized holothurians. Finally, we add the AFF module to realize the deep fusion of the shallow-detail and high-level semantic features of holothurians. The results show that the method presented in this paper yields better results on the 2020 CURPC underwater target-detection image dataset with an AP50 of 83.43%, Params of 15.90 M, and FLOPs of 25.12 G compared to other methods. In the underwater holothurian-detection task, this method improves the accuracy of detecting holothurians with fuzzy features, a small size, and dense scene. It also achieves a good balance between detection accuracy, Params, and FLOPs, and is suitable for underwater holothurian detection in most situations.

## 1. Introduction

Holothurians are nutritious and delicious, and they are loved around the world, especially in Asia. At present, due to the immaturity of technology, the harvesting of holothurian is still performed mainly by artificial diving, and intelligent holothurian-fishing robots have not been widely used. Fishing holothurians by hand is not only time-consuming, inefficient, and costly, but also poses a serious threat to divers’ lives. Therefore, the popularization of holothurian-fishing robots is still an inevitable trend. However, the holothurian-fishing robot must first solve the technological difficulty that identifies and locates the holothurians. Therefore, the underwater holothurian target-detection method is an urgent research subject that has great practical value.

Compared with terrestrial environmental target-detection methods, underwater target-detection methods face further challenges. Due to the complicated seabed environment and the limitations of imaging equipment, underwater images have noise pollution, low contrast, and color distortion. These problems seriously affect the performance of underwater biological target detection. The similarity in body color and the environment also makes the accurate detection of holothurians more difficult.

At present, underwater biological target-detection methods are divided into traditional and deep learning methods, and there is little research on holothurian-detection methods.

Traditional methods are based on the color, texture, and body edges of the subjects to identify the target. In 2012, Schoening et al. [1] proposed an automated detection method, iSIS (intelligent screening of underwater image sequences) to assist in the detection and monitoring of deep-sea benthic organisms, breaking new ground in marine research. In 2013, Fabic et al. [2] used the Canny edge test to extract fish contours during a fish-population-estimation mission. In 2014, Hsiao et al. [3], based on a sparse representation-based classification, proposed a partial-ranking method, SRC-MP, for underwater-video fish identification and for distinguishing swimming fish from other moving objects. In 2017, Qiao et al. [4] used a controlled limited adaptive histogram equalization (CLAHE) method for image processing to increase the contrast between holothurian spines and the body of the holothurian. The contours of the holothurian can then be accurately identified using edge-detection algorithms. In 2019, Qiao et al. [5] proposed an underwater holothurian-recognition method for support vector machines (SVMs). However, this method requires the texture features of the holothurian image dataset to be extracted first, and is susceptible to light conditions.

In recent years, more and more underwater target-detection methods based on deep learning have been proposed, most of which are improved versions of general target-detection models, which take into account the characteristics of the underwater environment. In 2015, Li et al. [6] used Fast R-CNN, then the latest targeted-detection method, to better effect fish-detection missions. In 2018, Martin Zurowietz et al. [7], in conjunction with Mask R-CNN, proposed machine learning-assisted image annotation (MAIA) to improve the annotation efficiency of large seafloor image sets. In 2020, Shi et al. [8] proposed the YOLOv3-marine algorithm for underwater target detection. YOLOv3-marine reduces network parameters and optimizes residual modules based on YOLOv3, resulting in improved detection accuracy and speed. In 2020, Liu et al. [9] proposed domain generalization YOLO (DG-YOLO) for underwater target detection. DG-YOLO is composed of YOLOv3, domain invariant module, and invariant risk-minimization penalty. In 2021, Zhang et al. [10] improved YOLOv4 by replacing the backbone network with a lighter MobileNet v2, while adding attention-mechanism modules that perform well in underwater target-detection missions. In 2022, Nils Piechaud et al. [11] used YOLOv4 for counting and measuring objects in the deep sea. In 2022, Lei et al. [12] proposed an improved YOLOv5 network for underwater target detection. By improving the multi-scale feature-fusion method of the YOLOv5 pathway aggregation network, the network can focus more on learning important features.

Compared with the underwater target-detection method, the common target-detection method based on deep learning on land is more advanced. At present, there are two kinds of common target-detection methods: the anchor-based and anchor-free methods. Classic methods of anchorage include Faster R-CNN [13], YOLOV3 [14], YOLOV4 [15], and YOLOV5. Although the anchor-based method is mature, there are always problems, such as an imbalance of positive and negative samples, memory consumption, and difficulty in identifying multi-scale targets. In order to solve these problems, more and more scholars have begun to study the anchor-free method, which is also the trend of the mainstream target-detection algorithm at present. Additionally, classic representatives of the anchor-free approach include CornerNet [16], ExtremeNet [17], CenterNet [18], and FCOS [19].

In order to achieve better holothurian detection, the anchor-free method, CenterNet, is used as the basic network in the current paper. For holothurian targets, we propose an improved detection algorithm for underwater holothurian targets based on CenterNet and scene feature fusion. This method can improve the detection accuracy for holothurians with fuzzy body features, small-sized holothurians, and high overlap, and can be deployed in embedded devices limited by resource requirements (for example, devices with low-graphics memory and low computational power). The main contributions of this paper are:(1)We propose an improved CenterNet model for holothurian detection that replaces the original backbone network, ResNet 50, with a more robust EfficientNet-B3. EfficientNet-B3 reduces the Params and FLOPs of the model, while increasing the depth and width of the model by using neural network architecture search (NAS) technology and the Depthwise Separable Convolution strategy. High-performance EfficientNet-B3 considerably improves the feasibility of deploying the model to resource-limited embedded devices.(2)In order to improve the accuracy of holothurian detection by making full use of the holothurian feature and co-existing scene information (e.g., waterweeds, reefs, and holothurian spines), we propose to add an FPT module between the backbone and neck networks. FPT uses three submodules, ST, GT, and RT, to integrate features from different scales and spaces, making full use of special scene features and details of holothurians to improve the accuracy of holothurian detection. At the same time, this paper improves the implementation of the FPT module in the target-detection network, adopts two FPT modules, inputs two different characteristic combinations, and then can integrate the model into more ecological scene information for holothurian detection.(3)In this paper, we use the AFF module to achieve a better integration of multi-scale features. Unlike conventional linear feature fusion (such as “Concat”), the AFF module can simultaneously combine global feature attention and local feature attention to achieve the effective fusion of low-level-detail and high-level-semantic features, thus improving the accuracy of holothurian detection.

Compared with other underwater target-detection methods, the FA-CenterNet (CenterNet+B3+FPT+AFF) method proposed in this paper achieves better detection accuracy on the CURPC underwater target-detection dataset. Additionally, the model’s FLOPs and Params are also controlled at lower values. The results also show that the proposed method can achieve a good balance between AP50, Params, and FLOPs, and validate the validity of our approach.

The rest of this paper is arranged as follows: in the second section, the structure of the underwater holothurian target-detection method FA-CenterNet is introduced in detail; in the third section, the experimental results and analysis are presented; and in the fourth section, the work of this thesis is summarized.

## 2. Proposed Method

### 2.1. Overall Network Structure

In this paper, the proposed method FA-CenterNet structure consisted of input, backbone, neck, head, and output networks. Figure 1 presents the overall network architecture of FA-CenterNet. In order to better complete the task of holothurian detection, the main improvements in this paper were to design and utilize the better lightweight network (EfficientNet-B3) as the backbone network of CenterNet, add two FPT (feature pyramid transformer) modules with different feature combinations between the backbone and neck networks, and use AFF modules to achieve feature fusion.

First, to make it easier for holothurian-detection models to be deployed in resource-limited embedded devices, we used EfficientNet-B3 as the backbone network in this paper. EfficientNet-B3, obtained using Google Neural Network Architecture (NAS) search techniques, has optimal model parameters, and the model contains a large number of Depthwise Separable Convolution and SE Attention Modules, enabling the model to perform well in terms of accuracy, Params, and FLOPs.

Then, neck network used the ConvTranspose operation to up-sample the multi-layer convolution results of EfficientNet-B3.

Then, in order to solve the difficulty of fuzzy features of holothurians, the FPT model was added to improve the detection accuracy of holothurians by using ecological scene information (e.g., waterweeds, reefs, and holothurian spines) that co-exist with holothurians. Additionally, we improved the implementation of FPT by using two FPT modules with different feature combinations to extract more holothurian scene information from the backbone network.

Then, feature fusion based on AFF modules was used between the backbone and neck networks. The AFF module can effectively integrate semantic and scale-incongruent holothurian features by enhancing channel attention between local and global features. Then, the features of the fusion were optimized again by using a 3 × 3 convolution.

Finally, the fusion features were introduced into CenterNet’s head module, and three independent head branches were used to generate critical point HEATmap, position offset, and target width. The final holothurian-detection result was obtained by a decoding operation.

In this paper, the loss function of FA-CenteNet was composed of three loss functions: heatmap, offset, and wh losses. We presented the specific formulas for the three loss functions in Appendix A.

### 2.2. EfficientNet-B3

In an underwater holothurian target-detection task, the trained model needs to be deployed into embedded equipment. However, the large computation volume and Params often bring great challenges to the embedded devices that are short of resources. Therefore, the lightweight nature of the model is very important for underwater holothurian target detection.

In this paper, we replaced CenterNet’s backbone network from ResNet50 to EfficientNet-B3 to address the model lightweight issues. EfficientNet [20] is a new lightweight network developed through Google’s Network Architecture Search (NAS) technology. In the ImageNet classification task, EfficientNet showed advanced performance in accuracy, FLOPs, and Params.

EfficientNet is guided by the idea that the model performs better by simultaneously scaling up its depth, width, and image resolution. A series of models of EfficientNet B0-B7 are obtained by scaling up the three dimensions of the model using different composite coefficients. EfficientNet-B3 has a high accuracy while maintaining smaller Params and less FLOPs, so we selected it as a feature extractor in this paper.

As shown in Figure 1 and Table 1, EfficientNet-B3 consists of one stem ordinary convolution layer and seven blocks. Blocks 1–7 are based on the MBConv module. There is a down-sampling relationship between blocks 2, 3, 5, and 7 in which stride is two. The default input resolution for EfficientNet-B3 was 300 × 300, and the proposed method in this paper adjusted it to 512 × 512.

MBConv mainly refers to MobileNet v2’s inverted residual structure and adds SE modules. The structure of MBConv is presented in Figure 2. In MBConv, when the expand ratio was 1, the input feature skipped the 1 × 1 convolution of the first layer and went directly to the Depthwise Separable Convolution module. When stride was two, the feature size shrank to 1/2 of its previous size. MBConv uses both the Depthwise Separable Convolution Strategy and SE Module, enabling EfficientNet-B3 to have an advanced performance in precision, Params, and FLOPs.

### 2.3. FPT Module

Holothurians have a good ability to protect themselves: their main body color can change with the environment’s color, so the body characteristics of holothurians and the environmental characteristics are highly similar. This is also the greatest challenge in holothurian-detection missions. As presented in Figure 3, the characteristic color of holothurian spines does not change with the color of the environment, but takes on a steady yellow-green cone shape. In addition, holothurian living environments generally have reefs, waterweeds, and other ecological scene information. These features from different sizes and spaces often exist in the same scene as holothurians, especially the small target holothurian spines from the lower networks. We believed that capturing the scene information was beneficial to the detection of fuzzy holothurians and will be a breakthrough to improve the detection performance of holothurians.

The SE module of the backbone network (EfficientNet-B3) continuously enhances inter-channel information by continuously obtaining the global feature weight between channels. However, as the number of layers deepens, the model gradually loses important local spatial details of holothurians (i.e., holothurian spines). To a certain extent, it affects the model’s ability to detect holothurians.

In order to solve the problem that the characteristics of holothurians are fuzzy and difficult to recognize, the FPT [21] (feature pyramid transformer) module was added to the proposed method. The FPT module incorporates features of holothurian ecological scenes (such as holothurian spines, reefs, and waterweeds) from different scales and spaces. These scene features can be used as auxiliary information to help the model detect holothurians. The FPT module is very useful for the recognition of fuzzy features and small-sized holothurians.

As presented in Figure 4, the input of the FPT module is a feature pyramid, and the output is a transformed feature pyramid that incorporates different levels of features. Compared with the classical feature pyramid network (FPN), the FPT module adopts a more complex feature-fusion strategy, so that each layer of the output has more contextual information. The FPT module is guided by transformers. It uses query (Q), key (K), and value (V) to capture contextual information, and then interacts with non-local features across space and scales to generate new feature maps.

The FPT module consisted of three types of transformers: a self-, grounding, and rendering transformers.

Unlike the original FPT fusion strategy, this paper used two FPT modules to fuse two feature combinations. This allowed the model to obtain richer feature information. The FPT fusion objects in this paper were (X0, X1, X2) and (X1, X2, X3), respectively.

As presented in Figure 4, the implementation details of FPT are described. First, feature combinations were processed by the transformers and the corresponding ST, GT, and RT features were obtained. Then, the new feature diagram was recombined to match the same sizes as X1 or X2. Finally, we used “Conv1” and “Conv2” to readjust the number of channels for the new feature-map combination, and then the final feature map was sent to the neck network.

As presented in Figure 5, to understand FPT in more detail, we obtained one of the FPT modules from the structure presented in Figure 4 as an example for detail analysis. The feature combination of the selected FPT models was (X1, X2, X3). X1, X2, and X3 represented the network’s high-level, mid-level, and low-level features, respectively. X3 details (e.g., obvious holothurian spines) interacted with X2 mid-level information (e.g., holothurian) to obtain the new feature RT (x2, x3). The holothurian features of X2 interacted with habitat features, such as reefs, to obtain a new feature, ST (X2). The high-level information of X1 (e.g., holothurians and ecological scenarios) interacted with the holothurians features of X2 to obtain the new feature GT (x2, x1). Then, three new features were combined with the original feature X2 to obtain the FPT output.

In this paper, the core idea of the FPT module was to obtain more features that co-exist with holothurians (e.g., waterweeds, reefs, and holothurian spines) as detection aids to improve the accuracy of the model for holothurians. Specifically, the FPT module integrates features from different scales and spaces through ST, GT, and RT components, providing greater weight to holothurians with unique scene and detail features, thus reducing the error-detection and missed rates for holothurians. The following is a detailed description of the FPT components (ST, GT, and RT).

#### 2.3.1. Self-Transformer

ST (self-transformer) [21,22] is a feature-interaction module based on non-local space, which can realize the information fusion of different spatial objects in the same scale feature map. In the underwater holothurian-detection task, ST can capture the relationship between holothurian and ecological scene features on the same scale. ST can use these scene features as auxiliary information for holothurian detection, and then enhance the model’s attention to this type of scene information, which improves the accuracy of holothurian detection. We presented the ST specific formula in Appendix B.

#### 2.3.2. Grounding Transformer

GT (grounding transformer) [21], as a feature-fusion module, uses semantic information at the top of the network to enhance the information at the middle and lower levels of the network. In the underwater holothurian-detection mission, GT can capture the relationship between the characteristics of holothurian ecological scenes (e.g., waterweeds, reefs, and holothurian spines) at different scales. Then, GT uses semantic information, such as large reefs in high-level networks, as an aid in detecting holothurians, and increased the model’s attention to holothurians in such scenarios. To a certain extent, GT improves the accuracy of holothurian detection with insufficient semantic information. We presented the GT specific formula in Appendix B.

#### 2.3.3. Rendering Transformer

RT (rendering transformer) [21], as a feature-fusion module, uses pixel-level information at the bottom of the network to render information at the middle and upper levels of the network. Unlike ST and RT, RT uses local spatial-feature interactions. Because the distance between non-local spatial features from different scales is too great, it makes little sense to capture the relationship between non-local spatial features. During the underwater holothurian-detection mission, RT enhanced the model’s attention to detail features, such as holothurian spines, thus improving the model’s accuracy in detecting holothurians with fuzzy body features. We presented the RT specific formula in Appendix B.

### 2.4. AFF Module

In general, the output of FPT is fused with the deconvolution module (neck) by means of “Concat”. However, this simple linear fusion is not the best way to integrate features that vary widely in semantics and scale. As presented in Figure 1, the proposed method replaces the original “Concat” feature fusion with an AFF (attentional feature fusion) module-based feature-fusion approach. This method can better integrate the features of holothurians with different semantics and scales, thus improving the accuracy of holothurian detection.

The core module of AFF [23] (attentional feature fusion) is MS-CAM and its structure is presented in Figure 6. Compared to SENet, MS-CAM not only extracts attention from global features, but also from channels with local features, with richer attention information. MS-CAM uses a combination of Global Branch + Local Branch. Unlike the local branch, the global branch adds an extra Global Avg Pooling. Global branching is used to extract attention from global features, helping models distinguish holothurian features from large-scale seafloor environments. The local branch uses pointwise convolution to extract channel attention for local features, which helps to enhance the model’s focus on the local features of holothurians (e.g., holothurian spines).

When AFF modules are used in feature pyramid structures, input feature X is the low-level detail feature of FPT output and input feature Y is the high-level semantic feature of the neck network. Based on MS-CAM, AFF’s computational equation can be expressed as follows:

(1)Z=M(X⊎Y)⊗X+(1−M(X⊎Y))⊗Y
where Z represents the output characteristics of the AFF module; M represents the MS-CAM module; ⊎ represents the initial feature integration; and ⊗ represents element by element multiplication.

Figure 7 presents the AFF module’s structure. The dotted line represents 1−M(X⊎Y). M(X⊎Y) and 1−M(X⊎Y) are both real numbers from 0 to 1. The advantage of AFF structural design is that the model can learn the weight between X and Y through its own training.

## 3. Experiments

### 3.1. Experimental Setting

#### 3.1.1. Dataset

We trained and validated our method on the CURPC 2020 dataset. Additionally, the CURP 2020 dataset is the official dataset provided by China’s underwater robot professional contest in 2020. There are 8200 images in the dataset, which included “holothurian”, “echinus”, “scallop”, and “starfish”. The dataset is in a VOC2007 format. Due to the large number of images in the CURPC dataset without holothurian targets, the quality of the model training was affected. Therefore, we read the XML file through the Python command and extracted the image containing the holothurians. In this work, 3333 images of holothurians and corresponding xml label files were extracted, including 1273 images at (3840, 2160) resolutions, 388 images at (1920, 1080) resolutions, 1671 images at (720, 405) resolutions and one image at (586, 480) resolutions. Additionally, there were 7214 holothurian targets in the 3333 images. The target holothurian in this dataset had the characteristics of multi-scale and overlapping, image blurring, low contrast, and occlusion, which made the target recognition more difficult. In order to demonstrate the robustness of the proposed method, the dataset image was the original real underwater image, without any image enhancement, closer to the real holothurian-fishing scene.

#### 3.1.2. Implementation Details

We trained our models on NVIDIA RTX 2080Ti. The experiment was based on Windows 10, Python 3.8, and Pytorch 1.2. 0. The specific configuration of the experiment is presented in Table 2.

**Training.** In this paper, the dataset was randomly divided into training and testing sets at a ratio of 9:1. In the data preprocessing stage, the proposed method in this paper, FA-CenterNet, was capable of the adaptive scaling of input images. The input image was set to a uniform size by grayscale filling to keep the holothurian features in true-body proportions. The size 512 × 512 was selected as the uniform input image size in this thesis. During training, the batch size was set at 6, epoch was set at 150, and the learning rate was set at 5 × 10^−4^. In addition, the optimizer was set to “Adam”, in which Adam’s mentum parameter was set to 0.9, and the learning-rate attenuation mode was set to “Cos”.

**Testing.** The confidence threshold was set at 0.3 and the nms iou used for non-extreme suppression was set at 0.3.

**Evaluation Metrics.** In this paper, we selected AP50, Params, and FLOPs as indicators to evaluate the model’s performance. Additionally, “AP50” means average precision when the IOU threshold is 0.5. “Params” means the number of trainable parameters in a deep network model. It measures the spatial complexity of model calculations in million (M), which can also be expressed as 1 × 10^6^. “FLOPs” is the floating point of operations. It measures the time complexity of the model calculations in giga (G), which can also be expressed as 1 × 10^9^. Params are related to the video memory of embedded devices, and the model with high Params has higher requirements for video memory; FLOPs are related to the chip computing power of embedded devices, and the model with high FLOPs has higher requirements for chips. Therefore, models with lower Params and FLOPs are easier to deploy in embedded devices.

### 3.2. Ablation Experiments and Analysis

#### 3.2.1. Quantitative Evaluations

To validate the efficacy of the FA-CenterNet model proposed in the current paper, we designed three ablation experiments to evaluate the effects of the backbone EfficientNet-B3 network, FPT module, and AFF module, respectively. First, the CenterNet with a backbone ResNet50 network was used as the baseline. Second, CenterNet’s backbone network was replaced with EfficientNet-B3 (named CenterNet (B3)). Third, on CenterNet (B3), add two FPT modules were added between the backbone and neck networks, incorporating contextual information between blocks 2, 3, 5, and 7 (named F-CenterNet). Fourth, we used the AFF module as a new feature-fusion method between FPT and neck based on F-CenterNet (named FA-CenterNet).

Table 3 presents the results of Centernet-based ablation experiments. The results show that the baseline model CenterNet has an AP50 of 79.03%, Params of 32.66 M, and FLOPs of 70.12 G.

Compared with CenterNet, CenterNet(B3) had an increase of 1.26% in AP50, a decrease in Params by 20.21 M, and an increase in FLOPs by 154.33 G. The results show that EfficientNet-B3 used as a backbone network significantly reduces the Params and FLOPs of the model. MBConv, the core module of EfficientNet-B3, employs two strategies: Depthwise Separable Convolution and SE Module. In addition, EfficientNet-B3 [20] determined the optimal parameter combination of the model using NAS search techniques. Therefore, EfficientNet-B3 can still provide greater accuracy and lower Params and FLOPs with a wider, deeper network structure than ResNet 50. The lightweight CenterNet model is easier to deploy on embedded devices.

Compared to CenterNet (B3), F-CenterNet increased by 2.58% on AP50, Params increased by 4.71 M, and FLOPs increased by 25.70 G. The results show that two FPT modules that incorporate different combinations of features can significantly improve the detection accuracy of holothurians, but also increase the model’s Params and FLOPs. The FPT module fuses holothurian features and scene features of different spaces and scales, which makes it suitable for the task of holothurian detection and achieves a superior performance in holothurian detection. In addition, F-CenterNet used “Concat” feature fusion between the backbone and neck networks, which led to an increase in model Params and FLOPs to some extent.

Compared to F-CenterNet, FA-CenterNet increased by 0.56% on AP50, Params decreased by 1.26 M, and FLOPs decreased by 16.18 G. The results show that the AFF module has a better fusion effect than the “Concat” mode. Although the FPT module incorporates rich contextual features, the output of FPT is still more skewed toward the lower layers of the network. It differs in semantics and scale from neck’s high-level features. The AFF module can simultaneously combine the attention of global and local features and realize the effective fusion of shallow and deep features of holothurians. Therefore, the AFF module improves the accuracy of the model for holothurian detection. In addition, the AFF module essentially adopts the “add” mode from the perspective of changes in feature spaces; compared to the “Concat” mode, the AFF module has less Params and FLOPs.

Overall, the method proposed in this paper (FA-CenterNet) improved by 4.40% in AP50, 16.76 M in Params, and 44.81 G in FLOPs compared to the original model (CenterNet). All three strategies, especially the FPT module, contributed to the model’s AP50 upgrade. The results show that the FPT module is very suitable for holothurian detection. Additionally, EfficientNet-B3, the backbone network, played an important role in the reduction in Params and FLOPs. Both FPT and AFF modules improved AP50 to varying degrees with the addition of only a few Params and FLOPs. Therefore, the results present the effectiveness of the three strategies in this paper.

#### 3.2.2. Effectiveness of Components in FPT

To validate the effectiveness of the three components of the FPT module in this paper, we designed a set of independent experiments for three transformer modules.

As presented in Table 4 of the results, ST, GT, and RT increases the accuracy of the model AP50 by 0.86%, 0.50%, and 1.36%, respectively, compared to CenterNet (B3). Of these, RT presented the most significant enhancement, suggesting that the fusion of more detailed information (such as holothurian spines) from lower levels of the network was most effective in improving the accuracy of holothurian detection. Finally, we observed that the FPT (a combination of ST, GT, and RT) suggested in this paper achieved the best detection accuracy, with AP50 increasing by 2.58% compared to the base model. The effectiveness of the three components of FPT and the FPT module proposed in this paper were been validated by this group of experiments.

#### 3.2.3. Impact of Score Thresholds

Figure 8 presents the precision, recall, and $F_{1}\text{-}score$ for four models of ablation experiments as the confidence thresholds change. The score threshold is also called the confidence threshold. The $F_{1}\text{-}score$ is a harmonic average of model precision and recall rates. Additionally, a higher $F_{1}\text{-}score$ means the model has a better performance. The calculation of the $F_{1}\text{-}score$ is presented in Equation (2):



(2)
$$F_{1}\text{-}score=2\times \frac{\;\mathrm{precision}\times \mathrm{recall}}{\;\mathrm{precision}+\mathrm{recall}}$$



From the precision and recall graphs, we can observe that when the score is between 0.1–0.4 and 0.3–0.8, respectively, the precision and recall are significantly different in the four models of the ablation experiment.

Compared to F-CenterNet(B3), F-CenterNet’s precision and recall were both significantly improved. It is shown that the FPT module made full use of holothurian spines, waterweeds, reefs, and other scene information, increased the model’s attention to the area with holothurian spines, and improved the accuracy of holothurian detection.

Compared to F-CenterNet, FA-CenterNet had a higher recall rate, but it also lost a small percentage of precision. It shows that the AFF module can not only integrate semantic and scale incongruities, but also cause some misdetection. Due to the complex environment of the seafloor and the poor imaging quality of underwater-camera equipment, underwater images have defects, such as noise pollution and fuzzy feature details, which affect the detection. Some of the waterweeds, reefs, and other features in the underwater images are similar to holothurians, as presented in Figure 9. AFF modules increase attention to the contextual features associated with holothurians through global and local attention. However, in the face of low-quality underwater images, AFF occasionally enhanced the attention to detail features similar to holothurians, resulting in a handful of false positives.

In the *F*_1_-*score* chart, all four models have the highest *F*_1_-*score* when the score is around 0.3. It can also be used as a basis for setting the confidence threshold in the model testing experiment so that the model can perform at its highest performance level. Overall, among them, FA-CenterNet had the best *F*_1_-*score*.

#### 3.2.4. Visualization of the Heatmap

Figure 10 visualizes the heatmap of four models in the ablation experiments. The brighter the color, the greater the weight.

Compared to CenterNet, CenterNet (B3) inhibited redundant background disturbance information on the seafloor, improving the accuracy of holothurian detection.

Compared to CenterNet (B3), the addition of F-CenterNet to the FPT module allowed the model to increase attention to the scene features associated with holothurians and improved the detection accuracy for holothurians with fuzzy features.

Compared to F-CenterNet, the FA-CenterNet, which adds an AFF attention module, was able to highlight holothurian targets more clearly with a smaller heatmap size.

By comparison, FA-CenterNet, which increased the FPT and AFF modules, significantly increased the model’s attention to fuzzy features, small size, and high-overlap holothurians at a lower Params and FLOPs cost. The visualization of the heatmap proves the validity of our method.

### 3.3. Comparison Experiments and Analysis

#### 3.3.1. Comparison with State-of-the-Art Methods

To validate the advantages of FA-CenterNet in underwater holothurian-detection missions, we evaluated our proposed approach in the CURPC 2020 dataset. Furthermore, FA-CenterNet was compared to SSD, YOLOv3, YOLOv4-tiny, YOLOv5-s, YOLOv5-l, and original CenterNet. Here, indicators, such as AP50, Params, and FLOPs, were used to evaluate the model’s performance. Specific results are presented in Table 5.

By comparing AP50, we found that the method proposed in this paper (FA-CenterNet) was superior to SSD, YOLOv3, YOLOv4-tiny, YOLOv5-s, and CenterNet. With the addition of the FPT and AFF modules, FA-CenterNet can make full use of holothurian scene features of different scales and spaces, effectively increasing the detection accuracy of holothurians with fuzzy features, small size, and high overlap. Additionally, FA-CenterNet’s AP50 reached 82.75%. In addition, YOLOv5-l had a great advantage in detection accuracy due to its deeper and wider network structure. FA-CenterNet achieved a detection accuracy similar to the heavyweight model YOLOv5-l, with a difference of only 0.71% compared to YOLOv5-l. YOLOV4-tiny’s AP50 was relatively low, 22.85% lower than FA-CenterNet’s. In order to improve the detection speed, YOLOV4-tiny only detected features at two scales in the final prediction phase, resulting in the low accuracy of YOLOV4-tiny in detecting small and overlapping targets.

By comparing Params, we observed that FA-CenterNet presented lower Params compared to YOLOv3, YOLOv5-l, and the original CenterNet. In these models, although YOLOV4-tiny had the best Params performance, with only 5.88 M, this was because YOLOv4-tiny significantly simplified the network structure, so the detection accuracy of the YOLOv4-tiny model decreased significantly, with AP50 being only 60.58%.

By comparing FLOPs, we observed that FA-CenterNet’s FLOPs were 25.12 M, which was 155.32 G lower than SSD and 90.8 G lower than YOLOv5-l, respectively. The results also show that FA-CenterNet has a clear advantage over FLOPs. Although YOLOv4-tiny and YOLOv5-s had the smaller FLOPs, they were at the expense of detection accuracy.

By combining AP50, Params, and FLOPs, we can conclude that the proposed FA-CenterNet achieved superior detection accuracy; simultaneously, it realized the balance between model lightweight and model accuracy.

#### 3.3.2. The Prediction Visualizations of Different Methods

Figure 11 presents the performance of different detection methods in the CURPC 2020 underwater target-detection dataset. Compared with other methods, the FA-CenterNet proposed in this paper successfully detected holothurians of different scales under water, with high detection accuracy and little leakage or misdetection.

Although SSD [24] adopts a pyramidal feature hierarchy structural strategy, there were too few network layers in the backbone network, resulting in too little semantic information in shallow networks. Therefore, the structural characteristics of SSD were not conducive to the detection of small-target holothurians, and the overall detection accuracy was low.

In pursuit of a lightweight model, YOLOV4-tiny uses only two scales of head to detect holothurian targets, resulting in a large number of smaller and overlapping holothurians being missed and significantly less accurate.

Overall, FA-CenterNet had a detection accuracy similar to the heavyweight model YOLOv5-l. We also observed that the FA-CenterNet proposed in this paper performed better than YOLOv5-l in dense, multi-scale holothurian scenarios due to its anchor-free mechanism.

## 4. Conclusions

At present, holothurians are still mainly caught manually. Due to noise pollution, low contrast, and color distortion in underwater images, the intelligent holothurian-fishing robot has encountered various technical difficulties in popularizing it. In order to avoid casualties in holothurian fishing, this paper proposed the method (FA-CenterNet) that combines high precision and its lightweight quality in order to promote the technological development of intelligent holothurian-fishing robots. The proposed method performed well in the CURPC 2020 datasets. Compared with other underwater target-detection methods, this method can improve the detection accuracy of fuzzy, small-sized, and overlapping holothurians. Meanwhile, the FA-CenterNet presents excellent performances of Params and FLOPs.

(1) In order to solve the problem of the resource limitation of embedded equipment, EfficientNet-B3 with its excellent performance was used as the backbone network. EfficientNet-B3 significantly reduced the model’s Params and FLOPs, making it easier for the model to be deployed in embedded devices, such as holothurian-fishing robots.

(2) In the current paper, the FPT module was added to deal with the difficulty of detecting holothurians better due to the complexity of the underwater environment and fuzzy features of holothurians. The FPT module could fully integrate the features of holothurian scenes (e.g., waterweeds, reefs, and holothurian spines) in different scales and spaces to improve the detection of holothurians with fuzzy features and highly similar bodies and backgrounds. The FPT module improved the implementation of the original FPT single-feature combination, and used two FPT modules as new fusion features in the model. Since the input had two different combinations of features, the model could be integrated into more ecological scene information for holothurian detection.

(3) In order to better integrate the different semantic features between the FPT output and neck-layer features, we proposed a feature-fusion method based on the AFF module. Compared with the “Concat” feature fusion in the conventional FPN structure, the AFF module simultaneously enhanced the model’s attention to global and local features, achieved the effective fusion of shallow and deep features, and improved the detection accuracy of holothurians.

(4) The method proposed in the current paper mainly identified and located underwater target detection of holothurians. The results show that FA-CenterNet has an AP50 of 83.43%, Params of 15.90 M, and FLOPs of 25.12 G on the CURPC 2020 underwater target-detection dataset. AP50 reflected the model’s ability to detect holothurians. Additionally, Params and FLOPs reflected the explicit memory space and chip computing power required by the model, respectively. Compared with other underwater target-detection methods, the proposed method, FA-CenterNet, achieved a good balance between detecting accuracy, Params, and FLOPs. FA-CenterNet can be used for real-world underwater holothurian-detection missions presenting an outstanding performance.

The method proposed in the current paper was helpful to promote the development of an intelligent holothurian-fishing robot and is of great significance to the further intelligent development of shallow-sea fisheries. The research in the future should focus on improving the model’s FPS performance by optimizing the model structure further to achieve a better balance of the model’s AP50, Params, FLOPs, and FPS performance metrics.

## Figures and Tables

**Figure 1 sensors-22-07204-f001:**
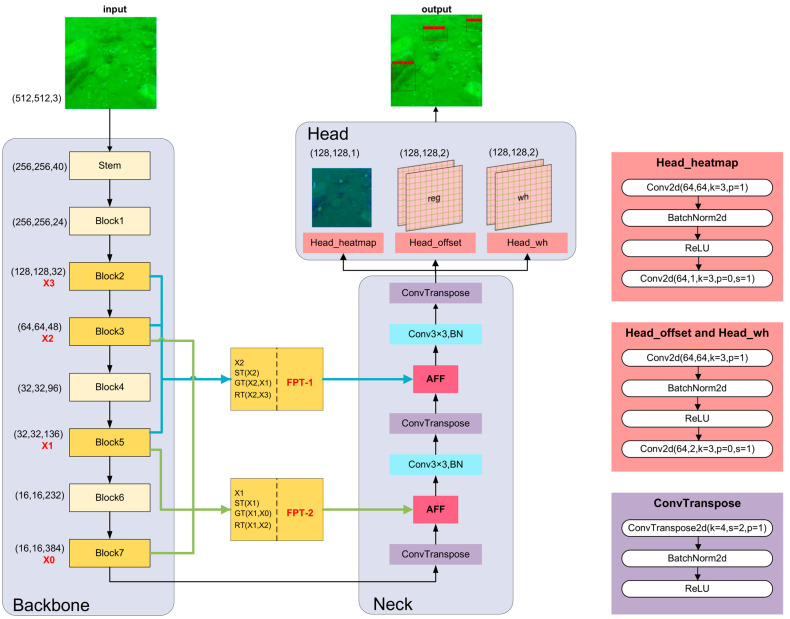
FA-CenterNet network structure. FA-CenterNet uses EfficientNet-B3 as its backbone network, adding FPT and AFF modules. Compared to the original CenterNet, FA-CenterNet improves the accuracy of underwater holothurian detection while reducing FLOPs and Params. There is a down-sampling relationship between blocks 7, 5, 3, and 2 in which stride is 2. For ease of describing the details of the FPT implementation, blocks 7, 5, 3, and 2 outputs are named X0, X1, X2, and X3. It can be observed that the FPT module incorporates two distinct sets of features.

**Figure 2 sensors-22-07204-f002:**
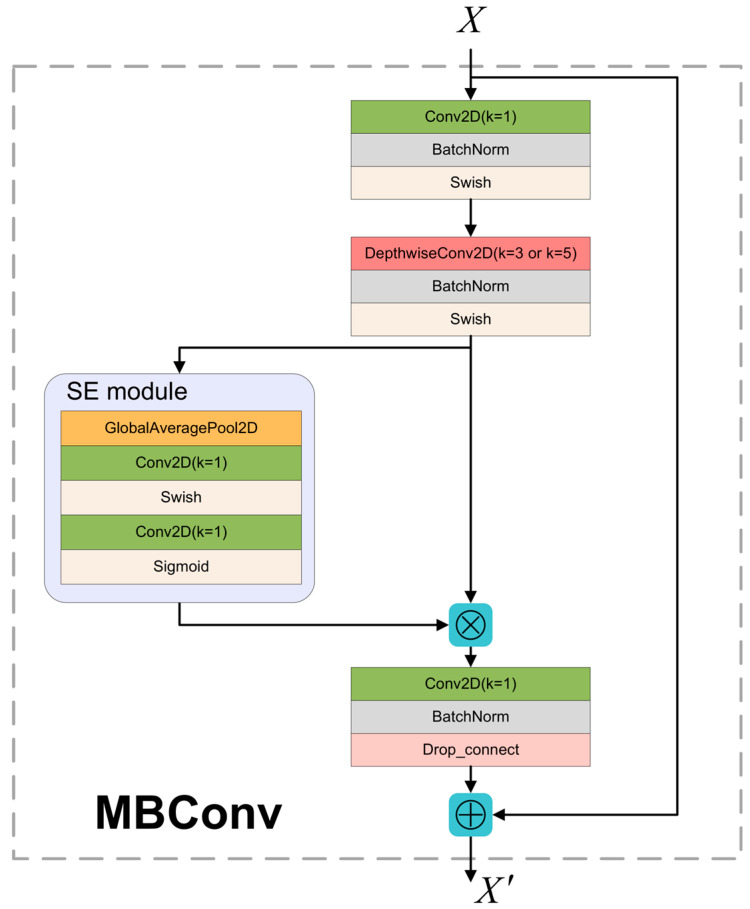
MBConv module.

**Figure 3 sensors-22-07204-f003:**
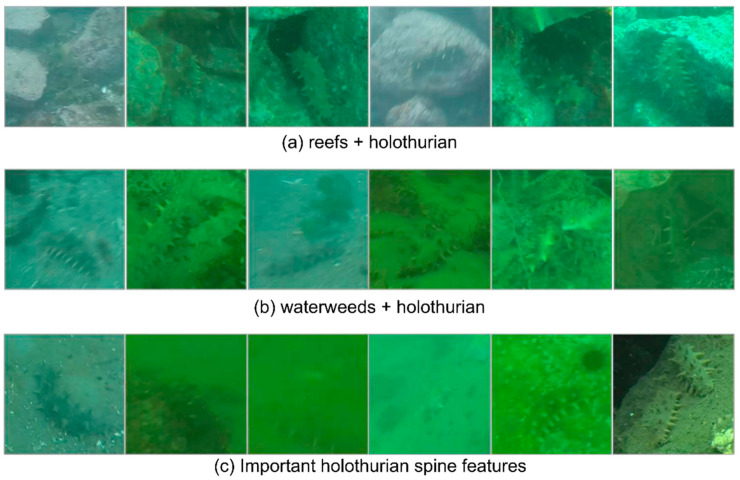
Holothurian scene in the CURPC dataset. (**a**) Reefs and holothurian appear in the same scenario. (**b**)Waterweeds and holothurian appear in the same scenario. (**c**) Holothurians whose body features are blurred but can be identified by its spines.

**Figure 4 sensors-22-07204-f004:**
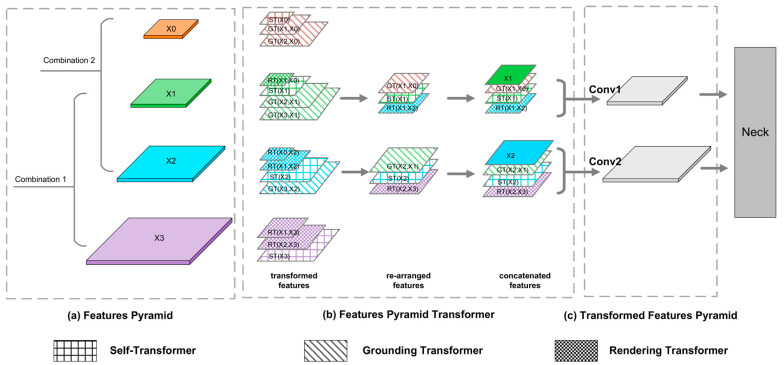
Improved structure of FPT modules. Different texture patterns represent different feature converters, and different colors represent feature maps of different scales. In order to describe the FPT module more succinctly, the outputs of blocks 7, 5, 3, and 2 are named X0, X1, X2, and X3. “Conv1” and “Conv2” on the right-hand side of the structure are 3 × 3 convolution modules with 192 and 96 output channels, respectively. (**a**) The FPT input is a feature pyramid consisting of two combinations. (**b**) FPT are the designs of three transformers (**c**) FPT output that controls the number of feature channels.

**Figure 5 sensors-22-07204-f005:**
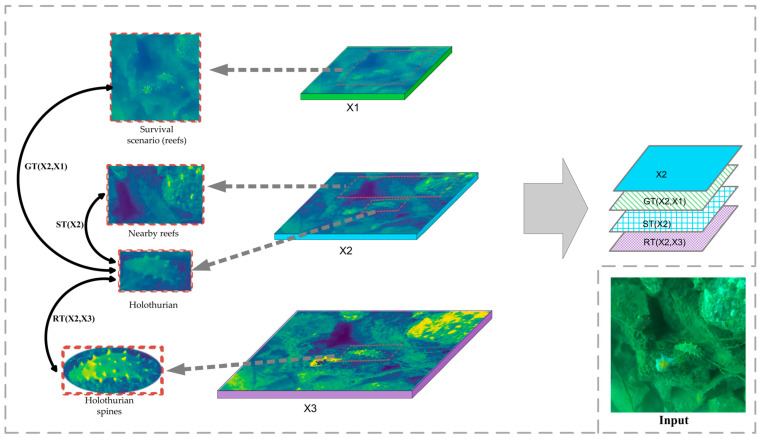
FPT feature interaction diagram.

**Figure 6 sensors-22-07204-f006:**
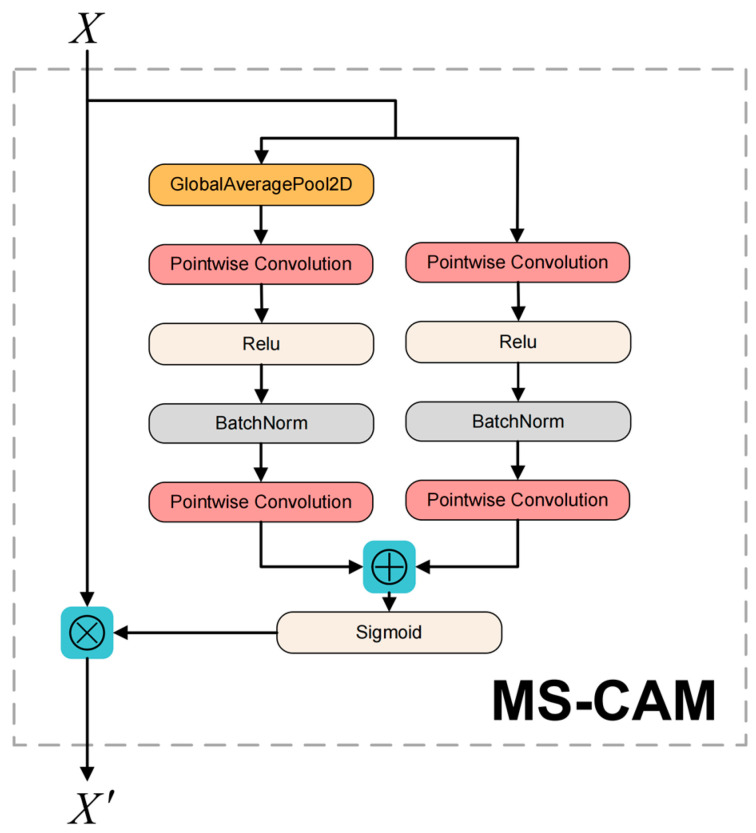
Structure of the MS-CAM module.

**Figure 7 sensors-22-07204-f007:**
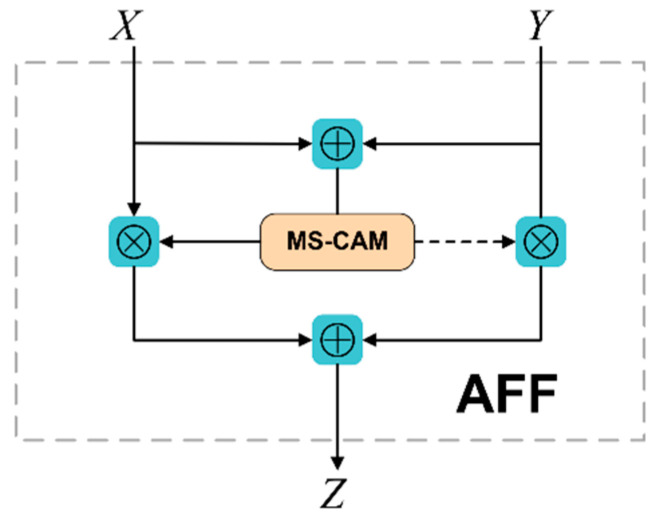
Structure of the AFF module.

**Figure 8 sensors-22-07204-f008:**
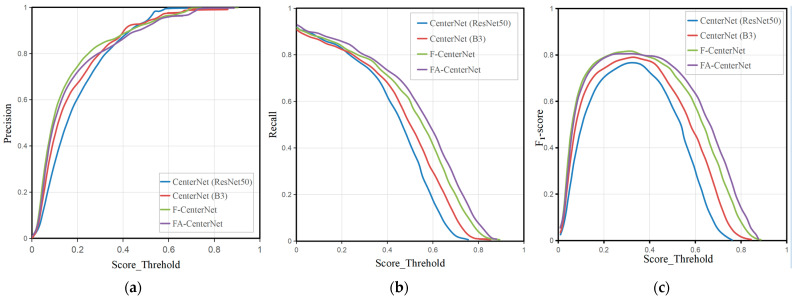
Impact of Score thresholds. (**a**) Precision vary with the score threshold. (**b**) Recall vary with the score threshold. (**c**) *F*_1_-*scores* vary with the score threshold.

**Figure 9 sensors-22-07204-f009:**
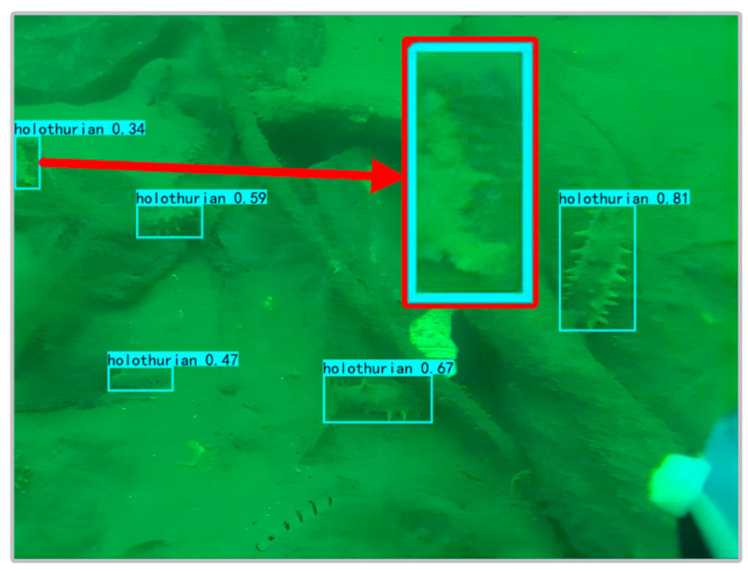
Waterweed falsely tested as holothurian.

**Figure 10 sensors-22-07204-f010:**
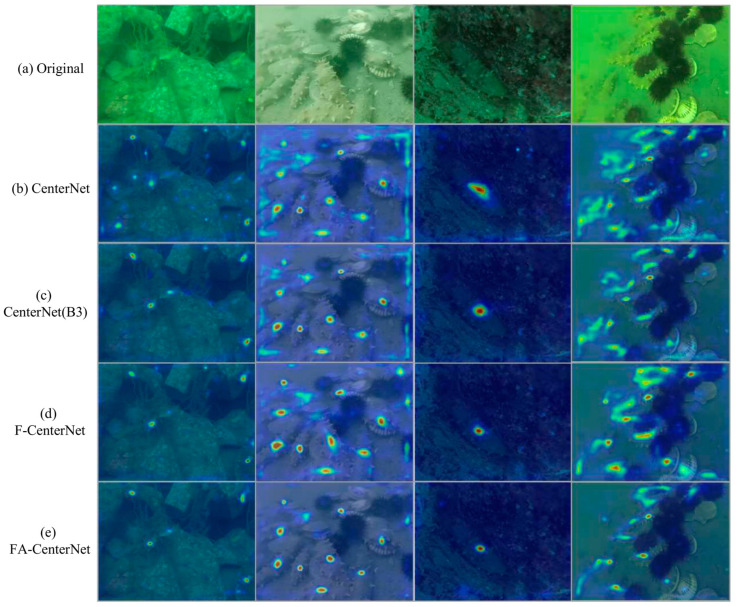
Visualizing heat maps of four models. (**a**) Original Input Pictures. (**b**) Heatmap visualization of the CenterNet. (**c**) Heatmap visualization of the CenterNet(B3). (**d**) Heatmap visualization of the F-CenterNet. (**e**) Heatmap visualization of the FA-CenterNet.

**Figure 11 sensors-22-07204-f011:**
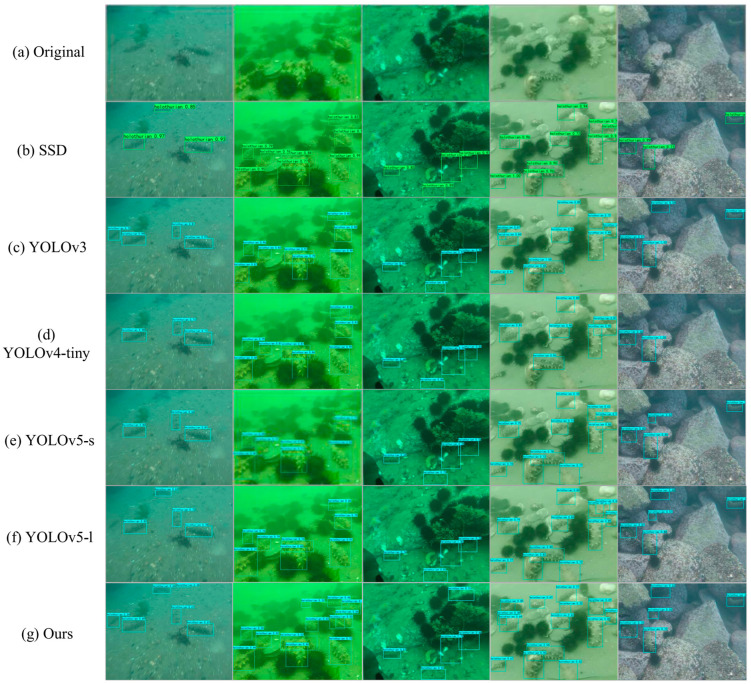
Performance of different detection methods in CURPC 2020 datasets. (**a**) Original Input Pictures. (**b**) Results of SSD. (**c**) Results of YOLOv3. (**d**) Results of YOLOv4-tiny. (**e**) Results of YOLOv5-s. (**f**) Results of YOLOv5-l. (**g**) Results of FA-CenterNet.

**Table 1 sensors-22-07204-t001:** Structure of EfficientNet-B3.

Module	Component	Component Layers	Kernel/Stride	Outputs
Stem	Conv	1	(3 × 3)/2	(256, 256, 40)
Block 1	MBConv1	2	(3 × 3)/1	(256, 256, 24)
Block 2	MBConv6	3	(3 × 3)/2	(128, 128, 32)
Block 3	MBConv6	3	(5 × 5)/2	(64, 64, 48)
Block 4	MBConv6	5	(3 × 3)/1	(32, 32, 96)
Block 5	MBConv6	5	(5 × 5)/2	(32, 32, 136)
Block 6	MBConv6	6	(5 × 5)/2	(16, 16, 232)
Block 7	MBConv6	2	(3 × 3)/1	(16, 16, 384)

**Table 2 sensors-22-07204-t002:** Experimental configurations used in this paper.

Environment	Version
CPU	Intel i9-10920X, 3.50 GHz
GPU	NVIDIA RTX 2080Ti
OS	Windows10
CUDA/CUDNN	V 10.1/V 7.6.5
Python	V 3.8
Pytorch	V 1.2.0

**Table 3 sensors-22-07204-t003:** Ablation experiment results.

Models	Backbone	FPT	AFF	AP50	Params	FLOPs
CenterNet	Resnet50			79.03%	32.66 M	70.12 G
CenterNet(B3)	EfficientNet-B3			80.29%	12.45 M	15.79 G
F-CenterNet	EfficientNet-B3	√		82.87%	17.16 M	41.49 G
FA-CenterNet	EfficientNet-B3	√	√	83.43%	15.90 M	25.31 G

**Table 4 sensors-22-07204-t004:** Impact of FPT components on model performance.

Models	Combination		AP50	Params	FLOPs
	FPT-1	FPT-2			
CenterNet(B3)	-		80.29%	12.45 M	15.79 G
+ST	X2, ST(X2)	X1, ST(X1)	81.15% (↑0.86%)	14.78 M	35.23 G
+GT	X2, GT(X2,X1)	X1, GT(X1,X0)	80.79% (↑0.50%)	16.67 M	34.94 G
+RT	X2, RT(X2,X3)	X1, RT(X1,X2)	81.65% (↑1.36%)	16.67 M	38.80 G
+FPT	X2, ST(X2), GT(X2,X1), RT(X2,X3)	X1, ST(X1), GT(X1,X0), RT(X1,X2)	82.87% (↑2.58%)	17.16 M	41.49 G

**Table 5 sensors-22-07204-t005:** Quantitative results of different detection methods in CURPC 2020 datasets.

Models	Backbone	AP50	Params	FLOPs
SSD	VGG19	76.30%	26.285 M	180.44 G
YOLOv3	DarkNet-53	75.03%	61.54 M	99.39 G
YOLOv4-tiny	CSPDarknet53-tiny	60.58%	5.88 M	10.34 G
YOLOv5-s	CSPDarknet	80.31%	7.07 M	10.56 G
YOLOv5-l	CSPDarknet	84.14%	47.01 M	115.92 G
CenterNet	ResNet50	79.03%	32.66 M	70.01 G
FA-CenterNet	EfficientNet-B3	83.43%	15.90 M	25.12 G

## Data Availability

Enquiries regarding the experimental data should be made by contacting the first author.

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
