# Peer review of "Underwater Holothurian Target-Detection Algorithm Based on Improved CenterNet and Scene Feature Fusion"

_sensors, 2022, doi:10.3390/s22197204_

Round 1
Reviewer 1 Report
The paper describes an underwater holothurian target detection algorithm, and the work is interesting, but some problems need to be resolved for the manuscript to be acceptable:
1.In lines 291-294, some details should be given, then it is easy to understand “the FPT module makes fully use of …………………………”
2.In line353-355,”Next, the GT uses semantic information such as large reefs in high-level networks as an aid in detecting …………”. It is suggested that some details should be given to prove that semantic information is used as aid in detection.
3.In line353-355, It is suggested that some data should be given to prove that” RT takes advantage of important holothurian spine features to improve the accuracy of the model for detecting holothurians with fuzzy body features”.
So I recommend to you that this manuscript should be revised carefully.
Reviewer 2 Report
The authors present a deep learning approach to the detection of holothuria / sea cucumbers in underwater image data. The approac h is based on applying three different strategies and a posterior extensive evaluation of the effects of these strategies. The authors can report a 4-5% increase in detection accuracy.
The main problem of the paper is the lack of focus on its main interesting findings. I would recommend to move some of the experiments and results and some details of methods to a supplementary file and concentrate on the main findings and observations. after reading the manuscript, a reader should understand what the main contribution of a paper is. This paper lists a large number of smaller and greater modification steps but if you are not a well trained deep learning enthusiast, the paper tells you almost nothing. The question is, are these adaptions only working on this CURPC 2020 data set so there is no greater value of these results for people not working with this data?
Another problem (which has something to do with the problem mentioned above) ist the list of references and related work. The authors do not really know, what kind of work is relevant : is it underwater computer vision for fauna detection (fishes, sea stars, coral, holothuria)? Then please cite other relevant work using deep learning like
1. Nils Piechaud, Kerry L. Howell, Fast and accurate mapping of fine scale abundance of a VME in the deep sea with computer vision, Ecological Informatics, 2022, 101786, ISSN 1574-9541,https://doi.org/10.1016/j.ecoinf.2022.101786
or
(2) MAIA—A machine learning assisted image annotation method for environmental monitoring and exploration. Martin Zurowietz, Daniel Langenkämper, Brett Hosking, Henry A. Ruhl, Tim W. Nattkemper, PLoS ONE, Nov 2018, https://doi.org/10.1371/journal.pone.0207498
Or are the authors only focussed on holothuria detection and also early works (liek the one from 2013 they are citing)? Then please cite
(3) Semi-automated image analysis for the assessment of megafaunal densities at the Artic deep-sea observatory HAUSGARTEN. T Schoening, M Bergmann, J Ontrup, J Taylor, J Dannheim, J Gutt, A Purser, TW Nattkemper, PLoS ONE, 2012, 7(6), DOI=10.1371/journal.pone.0038179
Anyway the paper is missing a reference to the CURPC 2020 data set and some references are incomplete (missing arxiv) like 7, 14, ....
Round 2
Reviewer 2 Report
The authors have addressed most imof my concerns. However I do not know os the darmta access is not sufficient, as the data (image/video plus labels) cannot be downloaded by anyone so nobody can reproduce these results. For me this is not acceptable.
But this must be decided by the editor.